# Scheduling Algorithm for Two-Workshop Production with the Time-Selective Strategy and Backtracking Strategy

Xiaohuan Zhang , Zhen Wang *, Dan Zhang and Tao Xu

School of Computer Science and Engineering, Huizhou University, Huizhou 516007, China
* Correspondence: wangzhen@hzu.edu.cn

**Abstract:** To solve the two-workshop integrated scheduling problem with the same device resources, existing algorithms pay attention to the horizontal parallel processing of the process tree and ignore the tightness between vertical serial processes. A scheduling algorithm for two-workshop production with the time-selective strategy and Backtracking Strategy is proposed. The scheduling order of each process in the process tree needs to be determined, which will be completed by the process sequence sequencing strategy. The scheduling plan also needs to be determined, which will be completed using the time-selective scheduling strategy for the two workshops. At the same time, the "reference time" is set for the current scheduling process. To find a better scheduling scheme, the "scheduling reference time" is recorded as T. If the time of the current scheduling process scheme is greater than T, the backtracking adjustment strategy will be used to track the process scheduling scheme. Finally, experiments show that the algorithm not only ensures the parallel processing of parallel processes but also effectively improves the tightness of serial processes and optimizes the results of integrated scheduling.

**Keywords:** scheduling algorithm; two-workshop production; time-selective strategy



## 1. Introduction

Collaborative manufacturing [1–3] can effectively integrate manufacturing resources and improve enterprises' flexible manufacturing [4–7] and anti-risk ability. In recent years, it has become a pressing issue for professional scholars. In studying the integrated scheduling problem of personalized products [8–12], distributed workshop collaborative manufacturing [13,14] is also performed. At present, there are few research achievements on this issue, and the main representative achievements are as follows:

(1) Ref. [15] effectively solves the problem of process parallelism in two workshops. However, it proposes to schedule leaf nodes first. This process will pay too much attention to the parallelism between processes and ignore the vertical scheduling of the process tree. Sorting processes only consider the processing time of the process and the batch of the process and do not consider the path length of the process. At the same time, it adopts the grouping process workshop determination strategy, which aims to reduce the migration times of processes in the second workshop. However, it ignores the path length where the process is located. In this way, most processes in the short path do not migrate, whereas a few processes in the long path migrate. The above results in the reduction of process tightness in the long path.

(2) Ref. [16] pays too much attention to the serial processing of the process tree and ignores parallelism. At the same time, a pre-scheduling strategy is proposed to make the end time of the two workshops close to each other. Its purpose is to balance the load of the second workshop and increase the parallel processing of processes. However, this method ignores the condition that when process migration occurs on a long path, the tightness between serial processes on the path will be reduced.

(3) Compared with the previous two methods, Ref. [17] pays more attention to the balance between the horizontal and vertical aspects of the process tree. However, it still does not consider some factors: 1. Parallel processes occupy processing equipment, which often leads to poor compactness between serial processes. 2. The impact of the first process on subsequent processes will lead to poor parallelism between processes. Therefore, it still has room for optimization. It proposes that the process groups with static overlapping time on the key devices be evenly distributed to the second workshop according to the scheme with the longest parallel time. Its purpose is to improve the parallelism of processes on crucial devices. However, this method ignores that the processes in the fixed overlapping time period do not necessarily overlap in the optimal scheduling scheme. Therefore, based on this, the processes on the key devices are allocated to the second workshop, which often leads to unnecessary process migration and preemption of processing devices and poor tightness of the serial process. The neighborhood rendering strategy and same device process workshop selection strategy are proposed to determine the process workshop. Based on the literature [16], the selection of a process workshop is considered carefully and integrated to reduce the number of process migrations. However, it still cannot effectively solve the unnecessary process migration and processing device preemption caused by the critical device balance strategy.

The above research considers the following problems:

(1) The scheduling algorithm has poor tightness of serial processes, which can still be optimized.
(2) The balance processing between two workshops should be paid attention to (e.g., process balance, load balance, and key device balance) while ignoring the balance. Processing is not the scheduling goal of the two workshops but a means to achieve the scheduling goal. In this method, the balance treatment is paid too much attention. This condition often leads to an increase in process migrations but affects the scheduling results.
(3) Reducing the number of process migrations should be paid attention to, ignoring the fact that the number of migrations is not always proportional to the total product processing time. In fact, in some cases, increasing the number of process migrations can often obtain more parallel time.

To solve the above problems, a two-workshop integrated scheduling algorithm based on time selection was proposed in [18]. The algorithm includes two strategies: the process sequence sorting strategy, which dynamically determines the processing sequence of each process from the overall structure of the process tree, and the workshop process sequence timing strategy, which generates several "trial scheduling schemes" for the scheduling process and selects the scheme closest to the scheduling target (i.e., the minimum total processing time) as the scheduling scheme of the process.

The algorithm no longer adopts a two-workshop equilibrium processing process. The reason is that the product processing process tree is evenly processed in the second workshop, which does not necessarily minimize the total production time. Therefore, forced balancing often leads to unnecessary process migration and device preemption, increasing the time overhead. The process migration time is used to replace the process migration times. When scheduling processes, the migration time of the previous process is considered, which intuitively reflects the impact of process migration on product scheduling. The algorithm can effectively improve the tightness between serial processes on the premise of ensuring parallelism. However, it still has disadvantages. There is a main problem in the process sequence timing algorithm. When selecting a scheduling scheme for each process, it only considers the minimum processing time in the current situation, but this does not mean overall optimization. This strategy is used to determine the process scheduling scheme, but it does not consider the subsequent scheduling process. Therefore, it still needs to be improved.

In this paper, an intelligent scheduling algorithm for two-workshop production with time-selective strategy is proposed. It proposes a backtracking adjustment strategy, which is based on the two-workshop integrated scheduling algorithm of a time-selective strategy. The algorithm is divided into two steps when determining the scheduling scheme of each process: step 1: To determine the process scheduling scheme. This is done according to the time-selective strategy of the process sequence; step 2: Set the "scheduling reference time" for each process scheduling scheme and compare it with the processing time of the current process scheduling scheme. Observe whether the processing time of the current process scheduling scheme is longer. If yes, the process scheduling scheme will be backtracked and adjusted according to the backtracking adjustment strategy. This is to find a better scheduling scheme in which the total processing time does not exceed the "scheduling reference time".

The rest of this paper is organized as follows: Section 2 shows the problem description and analysis. Section 3 discusses the strategy design. Section 4 shows the sample analysis. Section 5 discusses the experimental comparison and analysis. Section 6 concludes the paper with a summary.

## 2. Problem Description and Analysis

### 2.1. Problem Description

The integrated scheduling problem is to study how to schedule when a product is in the production mode of simultaneous processing and assembly, in which the process can minimize the completion time of the product. The product processing process diagram of the integrated scheduling problem presents a tree structure. Each node in the process tree represents the processing process, and an edge represents the partial order relationship of process constraints. The leaf node is the initial process, and the root node is the final processing process. It is assumed that workshop S1 and workshop S2 have the same device resources, and that there is a certain distance between them. The processes in the two workshops are processing and assembly. Here, processing and assembly are collectively referred to as processing. The processing device and assembly device are collectively referred to as processing devices, and the problems must meet the following conditions:

(1) Each device can only process one process at a time.
(2) Once the process begins processing, it is uninterrupted until the end.
(3) The processing is carried out in strict accordance with the partial order relationship in the process tree.
(4) Allow waiting between processes and allow the device to be idle before the process arrives.
(5) Each process has a unique processing device in each workshop.
(6) The process can be processed in a sure workshop.
(7) If the process is not in the same workshop as its immediately preceding process, the immediately preceding process shall be transferred before its processing.
(8) The total processing time of the product is the difference between the processing end time of the latest processing process and the processing start time of the earliest processing process.

**Definition 1.** *Product process tree: It describes the processing (assembly) processes in a product and the process constraint relationship between them. Because assembly needs to be considered, there is a many-to-one relationship.*

**Definition 2.** *Process sequence: It is a process set in which only a serial relationship exists between processes. Each process has at most one immediately preceding process and one immediately preceding subsequent process in the set.*

**Definition 3.** *Process queue: It stores the processes sorted by the process sequence sorting strategy.*

**Definition 4.** *Initial scheduling scheme: Only the scheduling scheme formed by the process on the longest process sequence in the product process tree is scheduled.*

**Definition 5.** *Process scheduling scheme: It refers to the scheduling scheme with process Wi (not the process on the longest process sequence) as the final scheduling process.*

**Definition 6.** *Scheduling reference time: The scheduling reference time of process Wi is the total processing time of its immediately preceding process Wi − 1 scheduling scheme Pi − 1.*

**Definition 7.** *Quasi-scheduling time point: Each alternative processing start time point of process Wi on its processing equipment is called the quasi-scheduling time point of process Wi.*

**Definition 8.** *Basic scheduling scheme: When scheduling the process Wi (not the last process in the longest process sequence), scheduling should be based on the process scheduling scheme Pi − 1 of the immediately preceding process Wi − 1 in its process queue. Then, scheme Pi − 1 is the basic scheduling scheme of process Wi.*

*2.2. Problem Analysis*

Suppose that the product consists of *num* processes. First, we sort the processing processes in the process tree. The scheduling is divided into two steps:

1. Find all processes on the longest process sequence, and schedule them to form an initial scheduling scheme (product scheduling scheme with n processes).
2. Each process is sequentially scheduled on the non-longest process sequence to form a process $Wi(n < I \leq Num)$ scheduling scheme $Pi(n < I \leq Num)$ (scheduling plan for some products containing $n + i$ processes). During this period, the time-selective strategy (including the time-selective scheduling strategy and time-selective adjustment strategy) and backtracking adjustment strategy will be used. In the production scheduling process, a total of $N + 1$ process scheduling schemes are formed. The $N + 1$ scheme is the product scheduling scheme, and its total processing time is the total processing time of the product.

The objective function and constraint function of the problem are shown in Equations (1)–(6). Objective function:

$$T = \min(Pn.tt) \tag{1}$$

$$Wi.at = Wi − 1.et + \alpha t \in \{0,1\} \tag{2}$$

$$Wi.st \geq Wj.at \tag{3}$$

$$P0 = \{W1, W2, \dots, Wn1\} \tag{4}$$

$$P1 = P0 \cup Wn1 + 1 \tag{5}$$

$$Pn1 = \{W1, W2, \dots, WN\} \tag{6}$$

$T$ is the total time of product processing; $Pn.tt$ is the total processing time of process $n2$ scheduling scheme; $Wi$ indicates the scheduling process; $Wi.at$ represents the arrival time of process $W$; $\alpha$ indicates whether process $Wi$ is migrated, $\alpha \in \{0,1\}$, where $\alpha = 1$ indicates migration and $\alpha = 0$ indicates that migration does not occur; $Wi.e.,t$ represents the processing end time of the process $Wi$; $Wi.st$ represents the processing start time of the process $Wi$; and $Wj$ is the immediately preceding process of process $Wi$. Equation (1) is the objective function of two job shop scheduling problems, which indicates that the total processing time (i.e., total product processing time) of the last process scheduling scheme should be the smallest. Equation (2) indicates that the arrival time of process $Wi$ is equal to the sum of the processing end time of process $Wi$-1 and its migration time. Equation (3) indicates that the processing start time of process $Wi$ is greater than or equal to the arrival time of its immediately preceding process. Equation (4) indicates that the initial scheduling scheme consists of the process scheduling on the longest process sequence.

Equation (5) shows that the scheduling scheme P1 of process $WN1 + 1$ is formed by the scheduling process $WN1 + 1$ on the basis of the process scheduling scheme P0. Equation (6) indicates that the final product scheduling scheme is the set of all process scheduling in the process tree.

## 3. Strategy Design

Three strategies were considered: process sequence sequencing strategy, initial scheduling scheme formation, and two-workshop time-selective strategy. They are exactly the same as those in [18], so they are not repeated here. Next, the above strategy was used to schedule the reverse-order product processing tree shown in Figure 1, and the disadvantages of the strategy in [18] were analyzed.

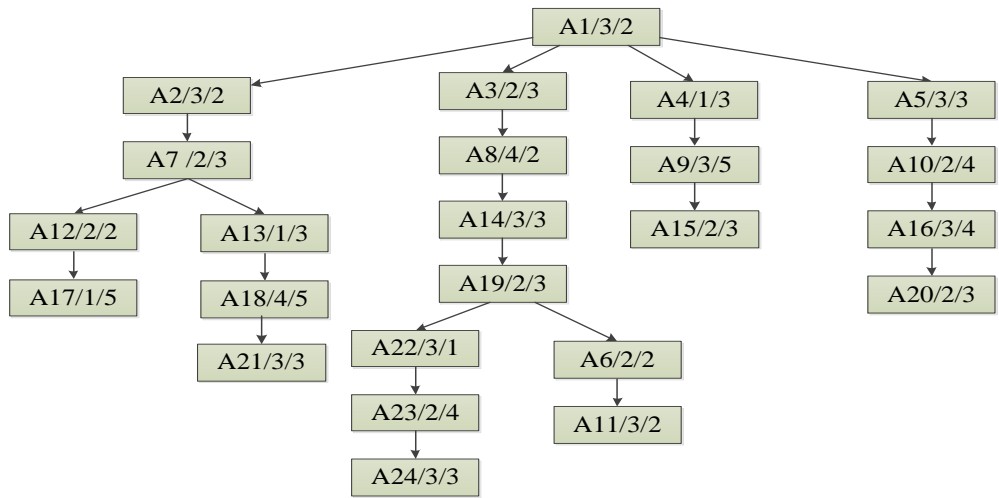

**Figure 1.** Product A processing process tree.

First, the process sequence sorting strategy was used to sort them. We calculated the path length of the current leaf node, and the results are as follows: A17:14, A21:18, A24:21, A11:17, A15:13, and A20:16. Therefore, the path length of A24 is the longest. We took all the processes on the path of A24 as the first process sequence (also the longest process sequence). These processes were inserted into the process queue line: A1, A3, A8, A14, a19, A22, A23, and A24. At the same time, they were deleted from the product process tree.

The product process tree that determines the longest process sequence is shown in Figure 2.

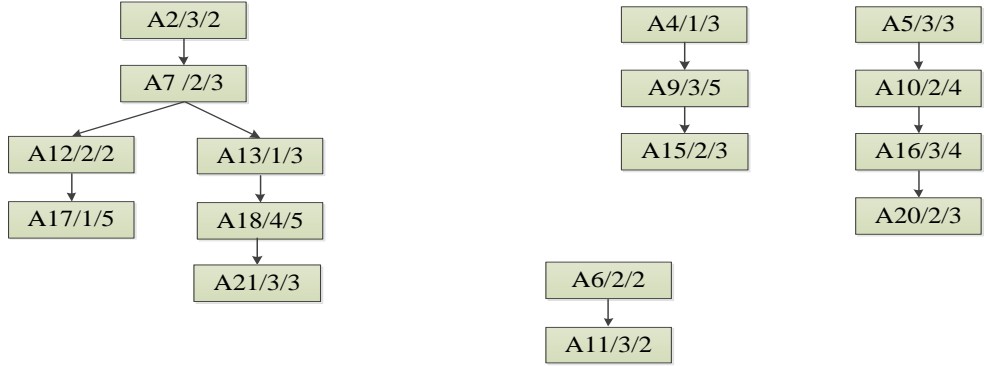

**Figure 2.** The product process tree becomes a forest when the first process sequence is determined.

Next, we calculated the path length of all leaf nodes in the current forest. The results are as follows: a17:12, a21:16, a11:4, a15:11, and a20:14. Therefore, we took all the processes on the path where A21 is located as the second process sequence and inserted these processes into the process queue in turn. At this time, the processes in the process queue are A1, A3,

A8, A14, A19, A22, A23, A24, A2, a7, A13, A18, and A21. At the same time, we deleted these processes in the product process tree.

According to the above methods, the sequence of subsequent processes shall be determined. The processes in the final product process queue are as follows: A1, A3, A8, A14, a19, A22, A23, A24, A2, a7, A13, A18, A21, A5, A10, a16, A20, A4, A9, A15, A12, A17, A6, and a11.

Then, the processes on the longest process sequence were scheduled to form the initial scheduling plan, as shown in Figure 3.

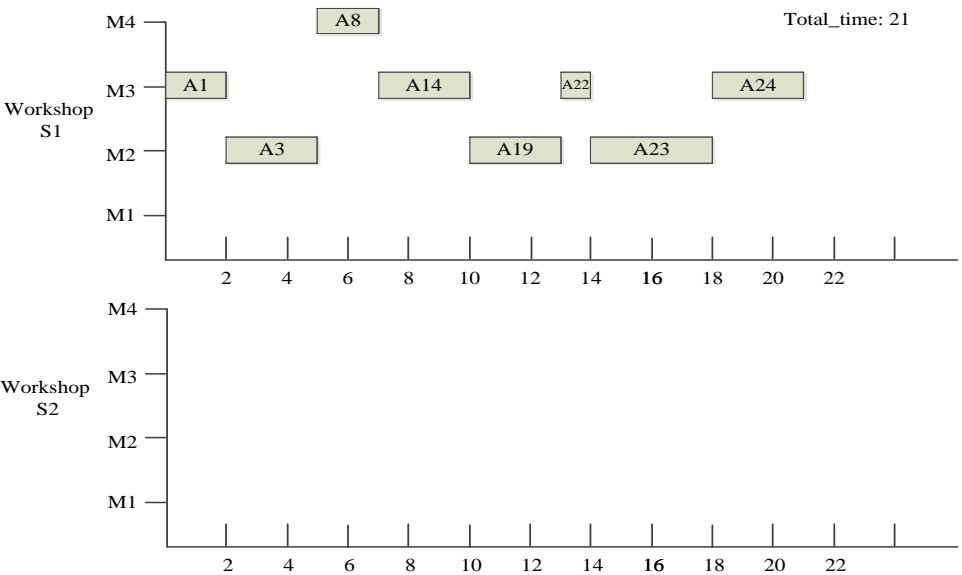

**Figure 3.** The initial scheduling scheme is formed by scheduling product A.

Next, the following process in the process queue is called A2. The processing device of process A2 is m3. Because it is a two-workshop scheduling problem with the same device resources, there are two M2 processing devices, one in workshop S1 and another in workshop S2. Based on the two-workshop timing algorithm, the trial scheduling time points of process A2 in workshops S1 and S2 are S1:2, S1:10, S1:14, S1,21, and S2:3. Among them, the total processing times of process A2 scheduling scheme obtained in the S1:2, S1:10, S1:14, S1:21, and S2:3 time point scheduling are 21, 21, 21, 23, and 21, respectively. We selected the process A2 scheduling scheme with the minimum total processing time. At this time, the total processing time of S1:2, S1:10, S1:14, and S2:3 is the minimum value of 21, indicating that the trial scheduling time point with the minimum total processing time is not unique. Then, we selected the time point with the earliest processing start time, that is, S1:2. The scheduling scheme of process A2 is shown in Figure 4.

We scheduled all subsequent processes in turn, and the final scheduling result is shown in Figure 5.

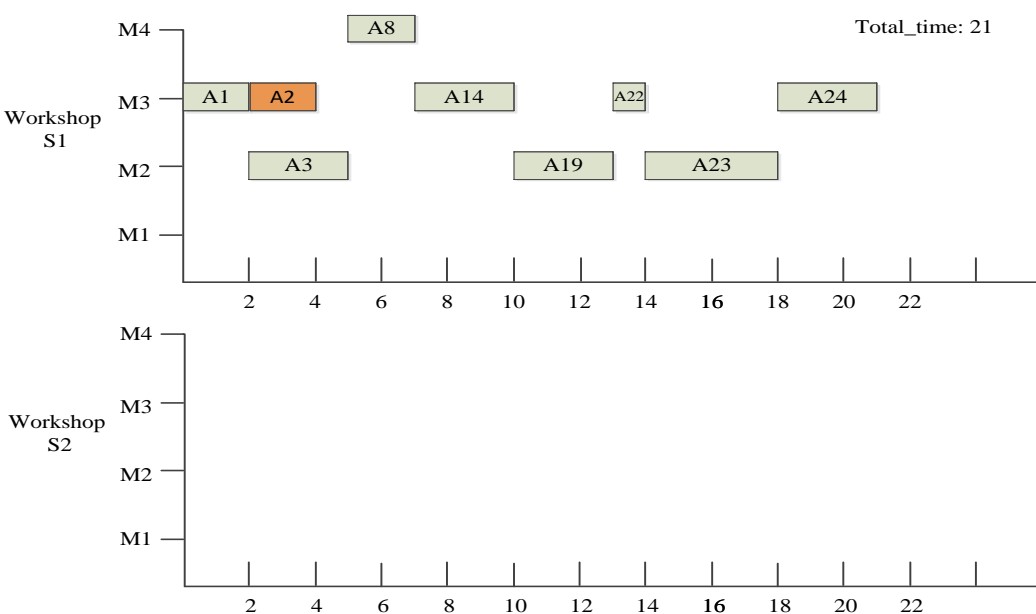

**Figure 4.** The process A2 scheduling scheme.

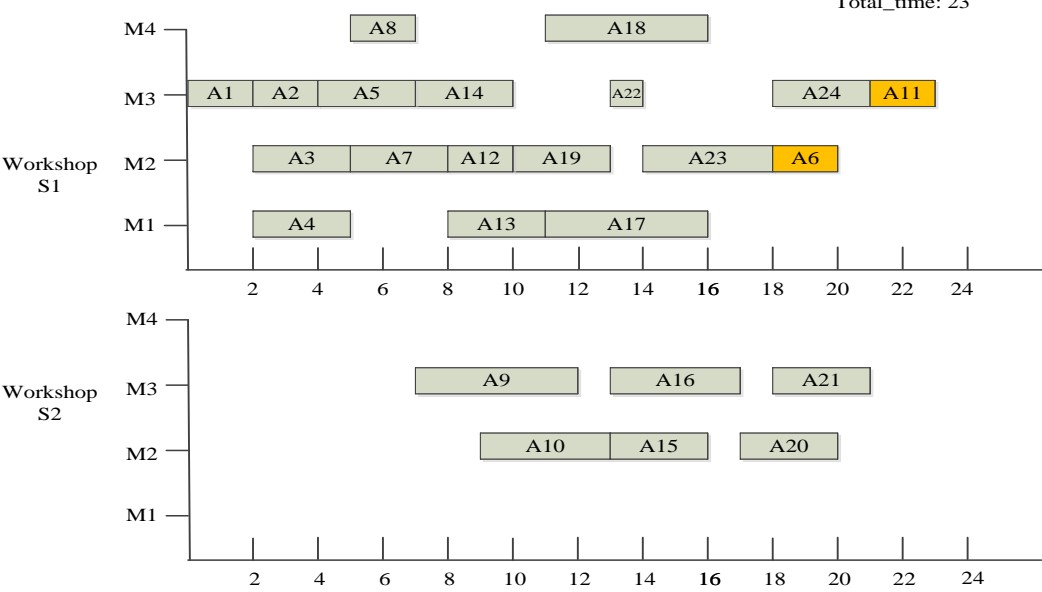

**Figure 5.** Gantt chart of algorithm scheduling results with time-selected strategy.

The analysis shows that when scheduling process A6, the timing algorithm selects the time point S1:18 because the scheduling degree A6 at this time point minimizes the current total processing time. However, on the whole, the scheduling of A6 affects the scheduling of A11 (It is marked orange in the Figure 5), resulting in unsatisfactory scheduling results. When scheduling A6, if you can select either S1:13 or S2:14, you will obtain better scheduling results. The current optimization does not represent the global optimization, and the two-workshop timing algorithm has disadvantages. Therefore, the backtracking adjustment strategy of the second workshop is designed to effectively optimize the results of the scheduling algorithm. The specific design ideas are as follows: Because the timing algorithm only considers the optimal scheduling results under the current scheduling state, the scheduling of subsequent processes is not considered. The second job shop backtracking scheduling strategy is further proposed based on the second job shop timing scheduling algorithm. First, a "scheduling reference time" is designed for each process to verify the process scheduling. A greater total processing time of the process scheduling scheme than

its "scheduling reference time" may be attributed to the influence of the scheduling of the previous process on the process scheduling. Therefore, the previous process needs to be rescheduled with the help of the backtracking adjustment strategy of the second workshop. At the same time, in each case of trial scheduling, we schedule the process and try to obtain a good solution by expanding the problem-solving space. When applying the backtracking adjustment strategy of the second workshop, two problems should be considered first. The first problem is when to start the second workshop backtracking adjustment strategy. For this problem, the "scheduling reference time" needs to be set for each process. The total processing time of the previous process scheduling scheme of the process will be used as the "scheduling reference time" of the process scheduling scheme. If the total time of the process scheduling scheme exceeds the preset "scheduling reference time," the scheduling of the previous process may affect the process, and the backtracking adjustment strategy of the second workshop is started. The second question is how many previous processes should be traced back. Obviously, the more the number of processes traced forward, the better the optimization effect of the algorithm. At the same time, the complexity of the algorithm increases exponentially. It can be set according to needs during application. The specific design scheme is as follows:

Set the backtracking parameters of the backtracking adjustment strategy of two workshops (i.e., the number of previous processes for the backtracking adjustment). Take backtracking parameter one as an example to introduce this strategy.

In the above example, the total processing time of the process scheduling scheme of process a11 exceeded its "scheduling reference time." Therefore, the backtracking adjustment strategy of the two workshops was enabled to reschedule its previous process A6. All the quasi-scheduling time points of A6 are as follows: s1:13, s1:18, s2:14, s2:15, and s2:19. We scheduled A6 at these time points, and then scheduled a11. The results are as follows: When A6 is scheduled at s1:13, a11 is scheduled at s1:15, and the total processing time of the a11 process scheduling scheme is 22. When A6 is scheduled at s1:18, a11 is scheduled at s1:21. The total processing time of the a11 process scheduling scheme is 23. When A6 is scheduled at s2:14, a11 is scheduled at s2:16. The total processing time of the a11 process scheduling scheme is 22. When A6 is scheduled at s2:15, a11 is scheduled at s2:20. The total processing time of the a11 process scheduling scheme is 23. When A6 is scheduled at s2:19, a11 is scheduled at s2:21. The total processing time of the a11 process scheduling scheme is 23. At this time, the scheduling time points with the minimum total processing time of the a11 process scheduling scheme are as follows: A6 at s1:13, a11 at s1:15, A6 at s2:14, and a11 at s2:16. We selected the time point with the earliest A6 processing start time, that is, A6 is at s1:13 and a11 is at s1:15 for scheduling. The total processing time of the a11 process scheduling scheme is less than the original. See Figure 6 for details.

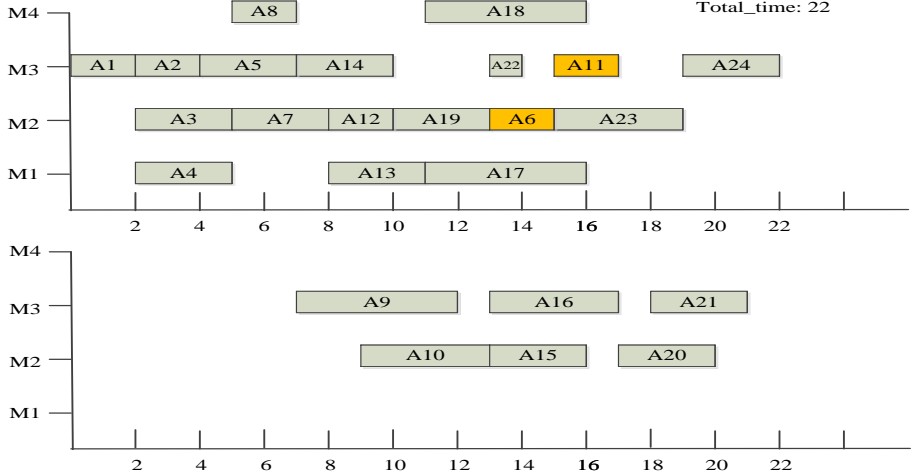

**Figure 6.** Gantt chart of scheduling results of the algorithm in this paper.

Figure 7 is the schematic diagram of the backtracking adjustment strategy of two workshops when the backtracking parameter is 1. In Figure 7, the total processing time of the process *i* scheduling scheme has exceeded its "scheduling reference time." The previous process of process *i* is process $(i - 1)$, so it is necessary to reschedule process $(i - 1)$. The basic scheduling scheme of process $(i - 1)$ needs to be determined, and the scheduling scheme of process $(i - 2)$ will be used as the basic scheduling scheme of process $(i - 1)$. Therefore, we found the quasi-scheduling time point of the process *i* on the process $(i - 2)$ scheduling scheme, scheduled process $(i - 1)$ on each quasi-scheduling time point, and obtained several process $(i - 1)$ quasi-scheduling schemes. Then, we found the quasi-scheduling time point of process *i* in the quasi-scheduling scheme of each process $(i - 1)$, scheduled process *i*, and obtained several quasi-scheduling schemes of process *i*. Finally, among several quasi-scheduling schemes of process *i*, the scheme with the lowest total processing time was used as the scheduling scheme of process *i*.

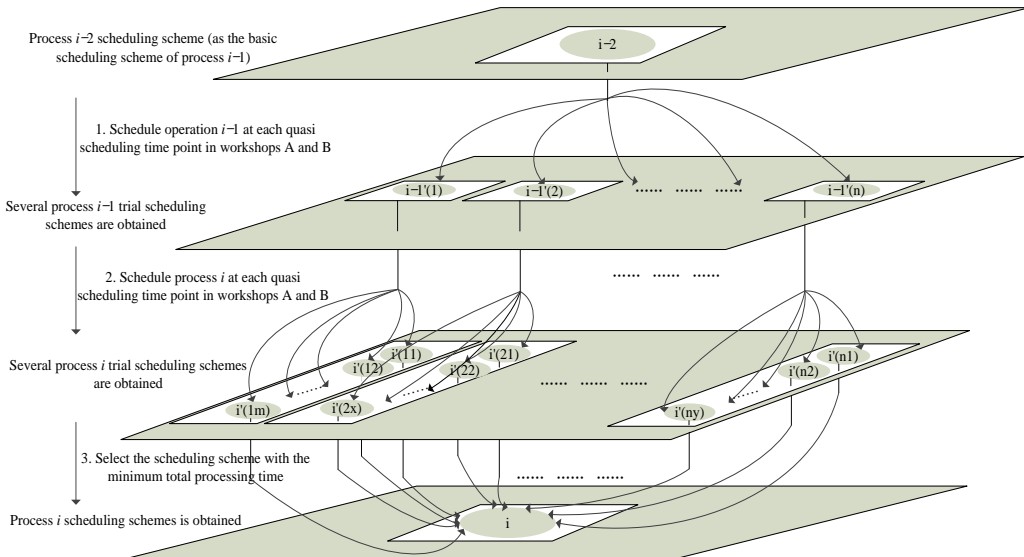

**Figure 7.** Schematic diagram of backtracking adjustment strategy of two-workshop when the backtracking parameter is one.

When the backtracking parameter is one, the steps of the backtracking adjustment strategy algorithm of the two workshops are as follows:

1.   Process *i* is set to the current scheduling. Judge whether the immediately preceding process $i - 1$ exists in the process sequence of process *i*. If it is, turn to 2; otherwise, turn to 13.
2.   Set the basic scheduling scheme of process $i - 1$ as scheme $Pi - 2$.
3.   Find the quasi-scheduling time points of process $i - 1$ in workshop *A* and workshop *B*, set a total of *M* ($M \geq 1$) quasi-scheduling time points, and use the two-workshop time-selective strategy to schedule process $i - 1$ on these quasi-scheduling time points to obtain m quasi-scheduling schemes of process $i - 1$.
4.   Set $K = 1$.
5.   Judge whether $K \leq m$. If yes, turn to 6; otherwise, turn to 9.
6.   Judge whether the total time of the *k*th quasi-scheduling scheme is greater than the total time of *Pi*. If yes, turn to 8; otherwise, turn to 7.
7.   Queue the scheme into *Q*.
8.   *K*++, go to 5.
9.   Set the total number of schemes in queue *Q* as *s*.
10.  Make a queue for queue *Q* and put the result in scheme *P*. If *P* is empty, go to 12; if *P* is not empty, go to 11.

11.  Take *P* as the basic scheduling scheme and use the second workshop time-selective strategy to schedule process i to obtain the proposed process i scheduling scheme *Pij* (1 ≤ *j* ≤ *s*), and turn to 10.

12.  Compare the newly obtained process *Pi* scheme with the previously obtained *Pi* scheme, select the scheme with the minimum total processing time, and determine it as the process *i* scheduling scheme, which is recorded as the process scheduling scheme *Pi*.

13.  Exit.

The backtracking adjustment strategy is the most complex part of the current algorithm, other operations are serial with it, and the time complexity is less than it. The algorithm time complexity of this strategy is O($n^2$). In the proposed algorithm, the number of operations using the backtracking adjustment strategy is (*n-N*), so the time complexity of the proposed algorithm should be this step requires O (*n-N*) × $n^2$, where *n* represents the number of processes in the process tree, and *N* represents the number of processes in the longest processes sequence.

## 4. Sample Analysis

Because the proposed algorithm is the result of the theoretical analysis and is not based on specific examples, it has universal significance. To help readers understand the algorithm, the proposed algorithm and the algorithm in [15–18] were applied to a single order of a manufacturing enterprise as an example to illustrate the advantages of the proposed algorithm. Product A consists of 24 processes and is processed on four devices. The reverse processing process tree of the product is shown in Figure 1. The number in each node of the process tree means the process number/processing device number/processing time. We used [15–18] algorithms and the proposed algorithm to schedule product A. When applying [15–17] to schedule products, the number of process migrations was not recorded. Instead, when the immediate process migration was caused by the scheduling of the process, the required migration time was arranged before the processing start time of process. That is, the processing start time of processmust be greater than or equal to the sum of the processing end time of its immediate process and its migration time. The dispatching Gantt chart is shown in Figures 5, 6 and 8–10. In each figure, the upper workshop is the S1 workshop, and the lower workshop is the S2 workshop.

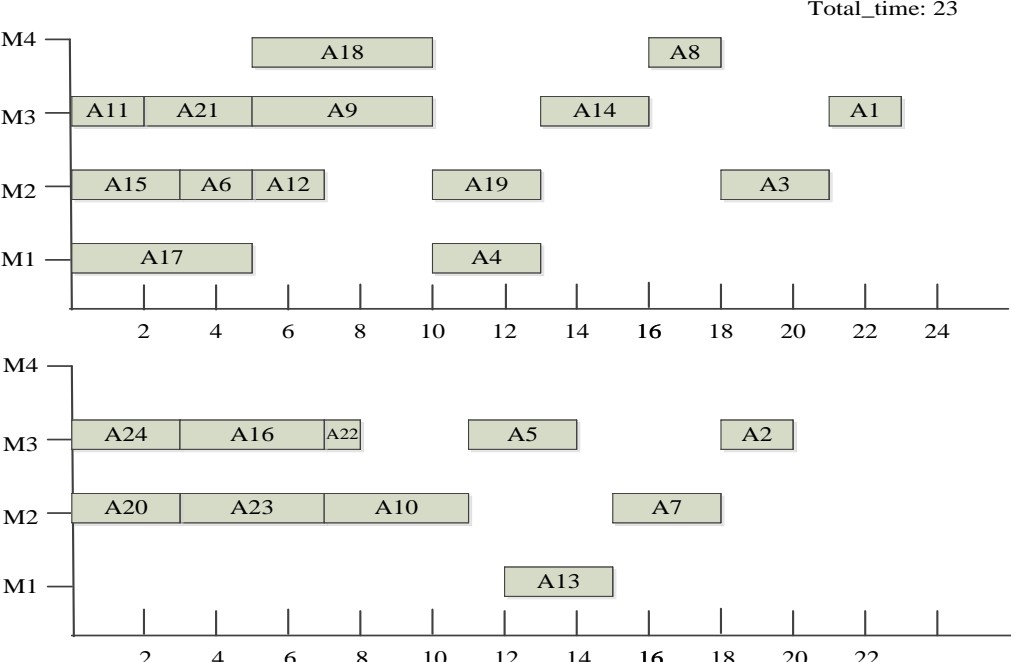

**Figure 8.** Gantt chart of algorithm scheduling results in the Ref. [15].

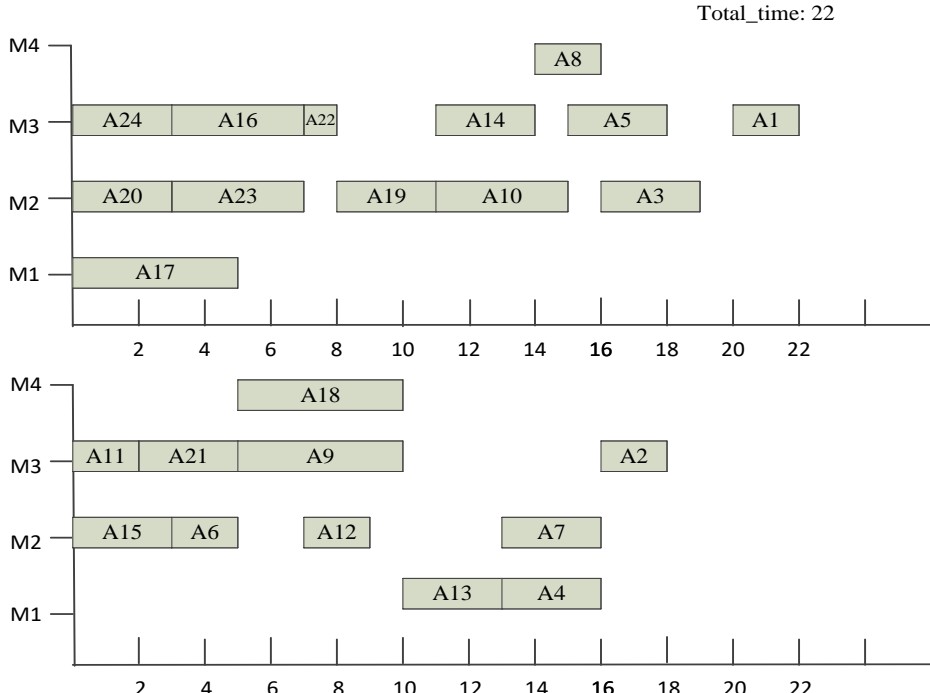

**Figure 9.** Gantt chart of algorithm scheduling results in the Ref. [16].

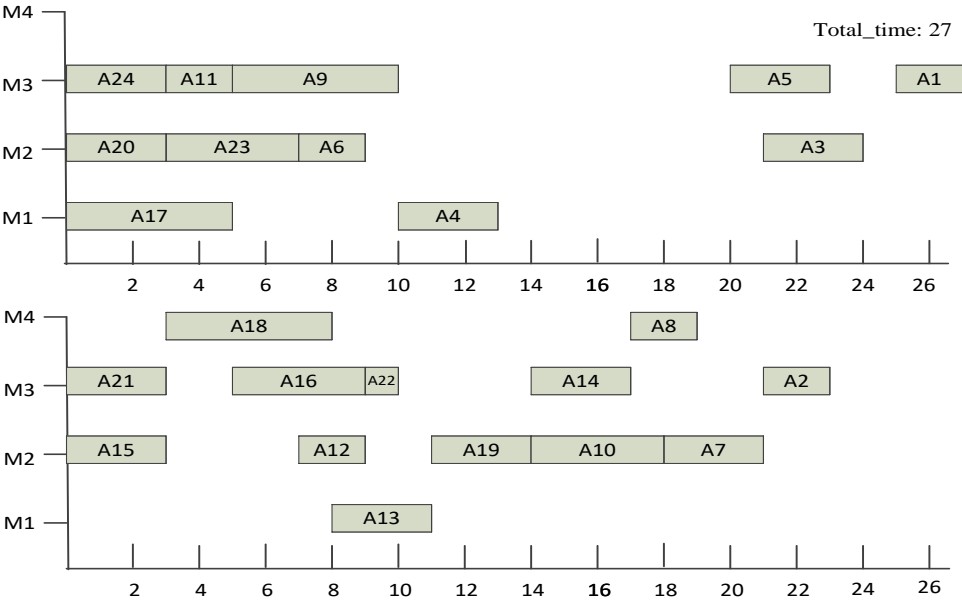

**Figure 10.** Gantt chart of algorithm scheduling results in the Ref. [17].

The experimental results show that the proposed algorithm is better than the above algorithm. Table 1 shows the scheduling process of the algorithm proposed in this paper.

**Table 1.** The scheduling process data of product A scheduled by the proposed algorithm.

| Id | Device | Quasi-Scheduling Time | Total Processing Time of Trial Scheduling Scheme | Determine the Scheduling Time | Determine the Processing Workshop | Is Backtracking Required | Each Process Scheduling Time in the Current Scheme |
|---|---|---|---|---|---|---|---|
| A1 | M3 | - | - | 0 | S1 | No | A1:0 |
| A3 | M2 | - | - | 2 | S1 | No | A1:0, A3:2 |
| A8 | M4 | - | - | 5 | S1 | No | A1:0, A3:2, A8:5 |
| A14 | M3 | - | - | 7 | S1 | No | A1:0, A3:2, A8:5, A14:7 |
| A19 | M2 | - | - | 10 | S1 | No | A1:0, A3:2, A8:5, A14:7, A19:10 |
| A22 | M3 | - | - | 13 | S1 | No | A1:0, A3:2, A8:5, A14:7, A19:10, A22:13 |
| A23 | M2 | - | - | 14 | S1 | No | A1:0, A3:2, A8:5, A14:7, A19:10, A22:13, A23:14 |
| A24 | M3 | - | - | 18 | S1 | No | A1:0, A3:2, A8:5, A14:7, A19:10, A22:13, A23:14, A24:18 |
| A2 | M3 | S1:2, S1:10, S1:14, S1:21, S2:3 | 21, 21, 21, 23, 21 | S1:2 | S1 | No | A1:0, A3:2, A8:5, A14:7, A19:10, A22:13, A23:14, A24:18, A2:2 |
| A7 | M2 | S1:5, S1:13, S1:18, S2:6 | 21, 23, 21, 21 | S1:5 | S1 | No | A1:0, A3:2, A8:5, A14:7, A19:10, A22:13, A23:14, A24:18, A2:2, A7:5 |
| A13 | M1 | S1:8, S2:9 | 21, 21 | S1:8 | S1 | No | A1:0, A3:2, A8:5, A14:7, A19:10, A22:13, A23:14, A24:18, A2:2, A7:5, A13:8 |
| A18 | M4 | S1:11, S 2:12 | 21, 21 | S1:11 | S1 | No | A1:0, A3:2, A8:5, A14:7, A19:10, A22:13, A23:14, A24:18, A2:2, A7:5, A13:8, A18:11 |
| A21 | M3 | S1:16, S2:17 | 22, 21 | S2:22 | S2 | No | A1:0, A3:2, A8:5, A14:7, A19:10, A22:13, A23:14, A24:18, A2:2, A7:5, A13:8, A18:11, A21:17 |
| A5 | M3 | S1:4, S1:10, S1:14, S1:21, S2:5 | 21, 21, 21, 24, 21 | S1:4 | S1 | No | A1:0, A3:2, A8:5, A14:7, A19:10, A22:13, A23:14, A24:18, A2:2, A7:5, A13:8, A18:11, A21:17, A5:4 |
| A10 | M2 | S1:7, S 1:8, S1:13, S1:18, S2:8, S2:20 | 25, 23, 23, 22, 21, 24 | S2:8 | S2 | No | A1:0, A3:2, A8:5, A14:7, A19:10, A22:13, A23:14, A24:18, A2:2, A7:5, A13:8, A18:11, A21:17, A5:4, A10:8 |
| A16 | M3 | S1:13, S1:14, S1:21, S2:12 | 25, 21, 25, 21, | S2:16 | S1 | No | A1:0, A3:2, A8:5, A14:7, A19:10, A22:13, A23:14, A24:18, A2:2, A7:5, A13:8, A18:11, A21:17, A5:4, A10:8, A16:12 |

**Table 1.** *Cont.*

| Id | Device | Quasi-Scheduling Time | Total Processing Time of Trial Scheduling Scheme | Determine the Scheduling Time | Determine the Processing Workshop | Is Backtracking Required | Each Process Scheduling Time in the Current Scheme |
|---|---|---|---|---|---|---|---|
| A20 | M2 | S1:17, S1:18, S2:16 | 26, 21, 21 | S1:18 | S1 | No | A1:0, A3:2, A8:5, A14:7, A19:10, A22:13, A23:14, A24:18, A2:2, A7:5, A13:8, A18:11, A21:17, A5:4, A10:8, A16:12, A20:16 |
| A4 | M1 | S1:2, S1:11, S2:3 | 21, 21, 21 | S1:2 | S1 | No | A1:0, A3:2, A8:5, A14:7, A19:10, A22:13, A23:14, A24:18, A2:2, A7:5, A13:8, A18:11, A21:17, A5:4, A10:8, A16:12, A20:16, A4:2 |
| A9 | M3 | S1:5, S1:7, S1:10, S1:14, S1:21, S2:6, S2:16, S2:20 | 30, 26, 23, 22, 26, 21, 24, 25 | S2:6 | S2 | No | A1:0, A3:2, A8:5, A14:7, A19:10, A22:13, A23:14, A24:18, A2:2, A7:5, A13:8, A18:11, A21:17, A5:4, A10:8, A16:12, A20:16, A4:2, A9:6 |
| A15 | M2 | S1:12, S1:13, S1:18, S2:11, S2:12, S2:18 | 26, 23, 21, 25, 21, 21 | S2:12 | S2 | No | A1:0, A3:2, A8:5, A14:7, A19:10, A22:13, A23:14, A24:18, A2:2, A7:5, A13:8, A18:11, A21:17, A5:4, A10:8, A16:12, A20:16, A4:2, A9:6, A15:12 |
| A12 | M2 | S1:8, S1:13, S1:18, S2:9, S2:12, S2:15, S2:18 | 21, 22, 21, 22, 21, 21, 21 | S1:8 | S1 | No | A1:0, A3:2, A8:5, A14:7, A19:10, A22:13, A23:14, A24:18, A2:2, A7:5, A13:8, A18:11, A21:17, A5:4, A10:8, A16:12, A20:16, A4:2, A9:6, A15:12, A12:8 |
| A17 | M1 | S1:10, S1:11, S2:11 | 23, 21, 21 | S1:11 | S1 | No | A1:0, A3:2, A8:5, A14:7, A19:10, A22:13, A23:14, A24:18, A2:2, A7:5, A13:8, A18:11, A21:17, A5:4, A10:8, A16:12, A20:16, A4:2, A9:6, A15:12, A12:8, A17:11 |
| A6 | M2 | S1:13, S1:18, S2:14, S2:15, S2:19 | 22, 21, 22, 23, 21 | S1:18 | S1 | No | A1:0, A3:2, A8:5, A14:7, A19:10, A22:13, A23:14, A24:18, A2:2, A7:5, A13:8, A18:11, A21:17, A5:4, A10:8, A16:12, A20:16, A4:2, A9:6, A15:12, A12:8, A17:11, A6:18 |
| A11 | M3 | S1:20, S1:21, S2:21 | 25, 23, 23 | S1:21<br>S1:13 | S1<br>S1 | Yes | A1:0, A3:2, A8:5, A14:7, A19:10, A22:13, A23:14, A24:18, A2:2, A7:5, A13:8, A18:11, A21:17, A5:4, A10:8, A16:12, A20:16, A4:2, A9:6, A15:12, A12:8, A17:11, A6:18, A11:21 |

**Table 1.** *Cont.*

| Id | Device | Quasi-Scheduling Time | Total Processing Time of Trial Scheduling Scheme | Determine the Scheduling Time | Determine the Processing Workshop | Is Backtracking Required | Each Process Scheduling Time in the Current Scheme |
|---|---|---|---|---|---|---|---|
| A6 | M2 | (A6 S1:13) A11 S1:15; (A6 S1:18) A11 S1:21; (A6 S2:14) A11 S2:16; (A6 S2:15) A11 S2:20; (A6 S2:19) A11 S2:21; | 22, 23, 22, 23, 23 | | | | A1:0, A3:2, A8:5, A14:7, A19:10, A22:13, A23:14, A24:18, A2:2, A7:5, A13:8, A18:11, A21:18, A5:4, A10:8, A16:12, A20:16, A4:2, A9:6, A15:12, A12:8, A17:11, A6:13 |

## 5. Experimental Comparison and Analysis

This section will verify the effectiveness of the algorithm. In the experiment, four groups of data were randomly generated in different data ranges. Data 1: the number of processes is 3–10, the processing time of each process is 1–6, and the number of devices is 3–5; Data 2: the number of processes is 10–20, the processing time of each process is 1–6, and the number of devices is 3–5; Data 3: the number of processes is 20–30, the processing time of each process is 1–6, and the number of devices is 3–5; Data 4: the number of processes is 30–40, the processing time of each process is 1–6, and the number of devices is 3–5. Each group of data will randomly generate 50 products. Because the product structure is random, it can effectively prove the algorithm's effectiveness in different cases. Then, the proposed algorithm (the backtracking parameters are 1, 2, and 5, respectively) is compared with [15–18], as shown in Figure 11, Figure 12, Figure 13, Figure 14, respectively. The comparison of the average scheduling results of each algorithm is shown in Figure 15.

It can be seen from the experimental results that the scheduling results of the algorithm proposed in this paper are significantly better than those in [15–18]. For the algorithm proposed in this paper, the larger the backtracking parameters, the better the scheduling results.

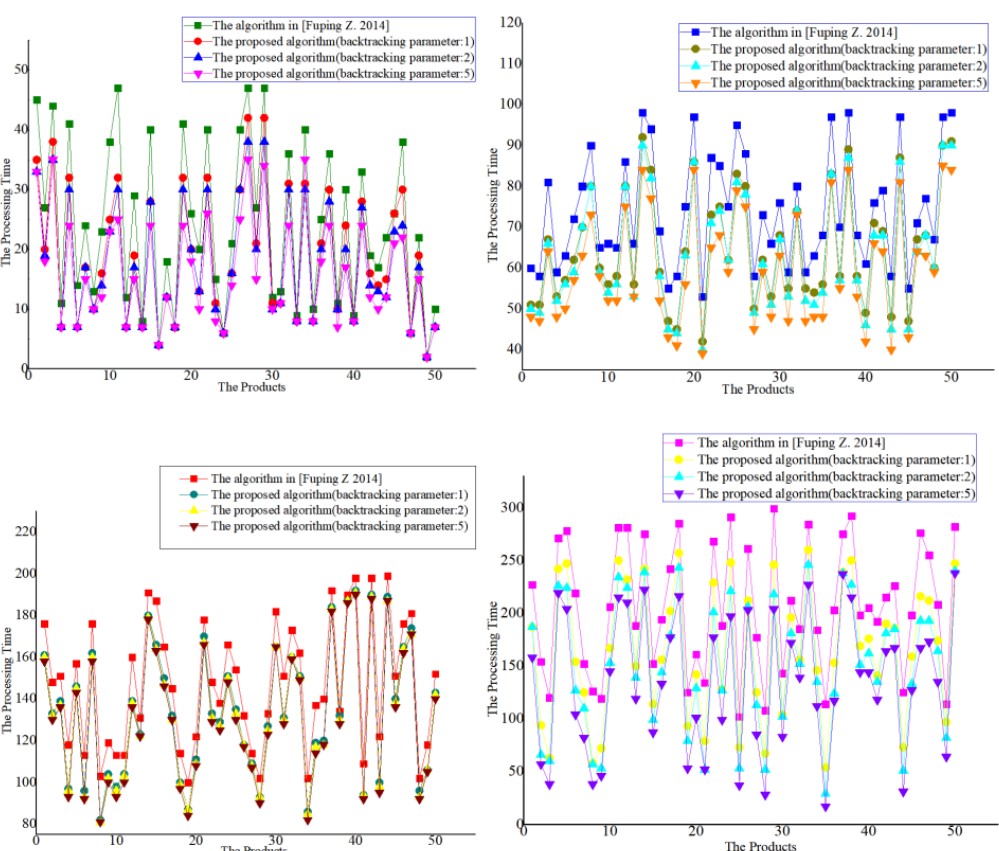

**Figure 11.** Comparison between the proposed algorithm and Ref. [15].

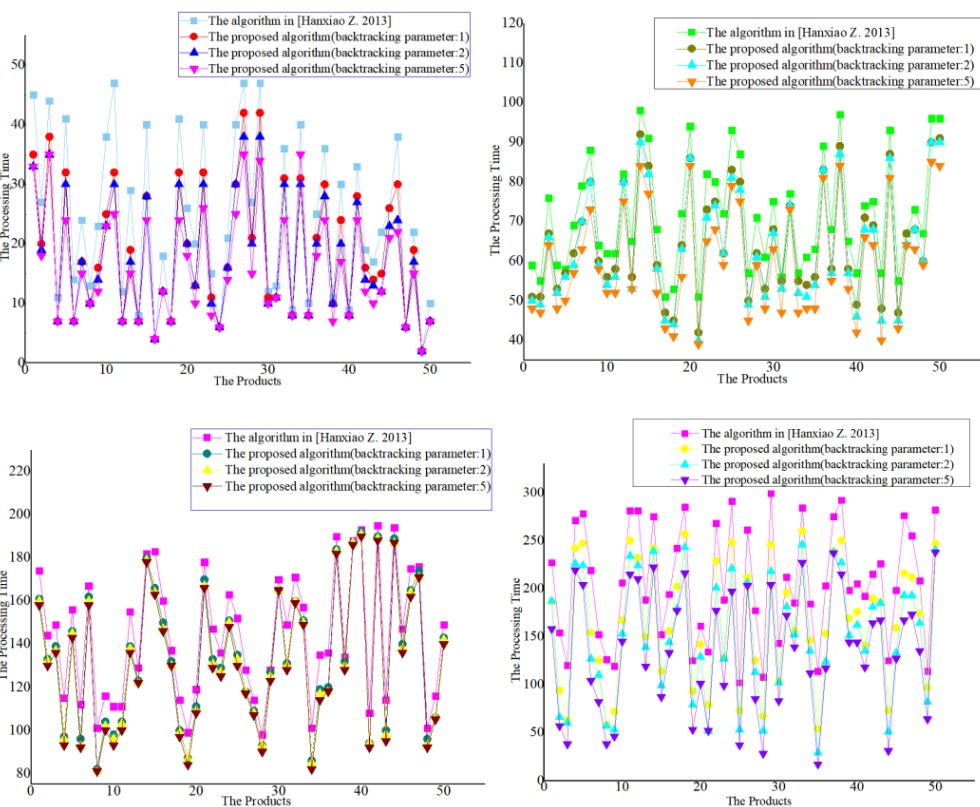

**Figure 12.** Comparison between the proposed algorithm and Ref. [16].

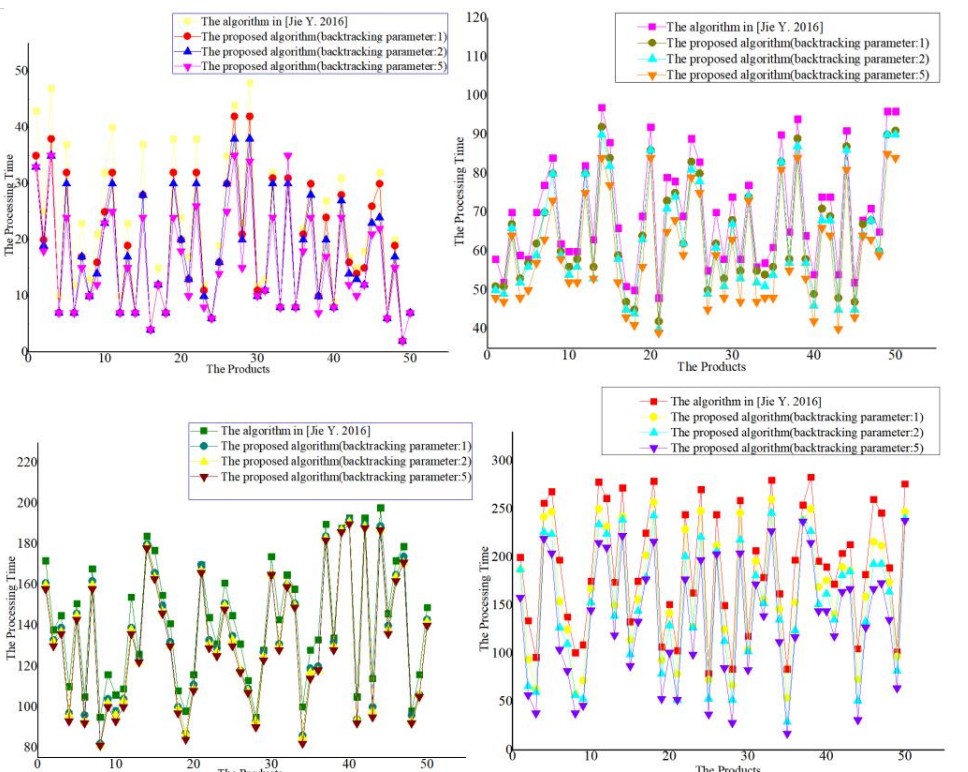

**Figure 13.** Comparison between the proposed algorithm and Ref. [17].

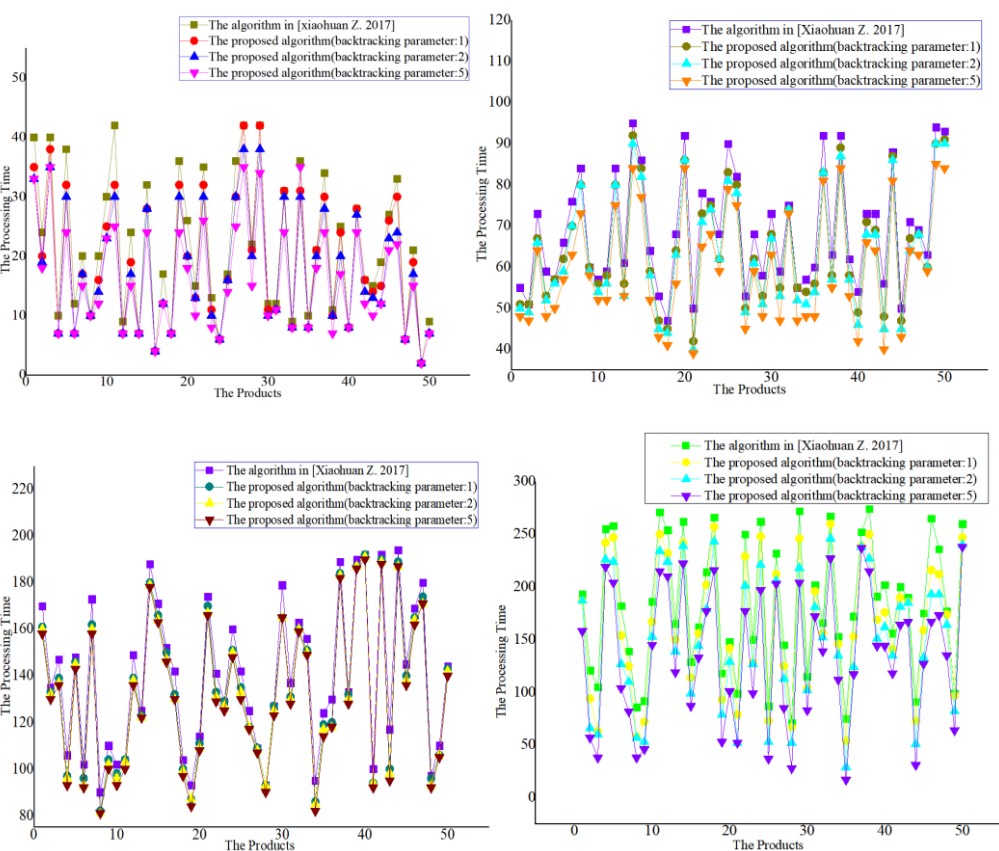

**Figure 14.** Comparison between the proposed algorithm and Ref. [18].

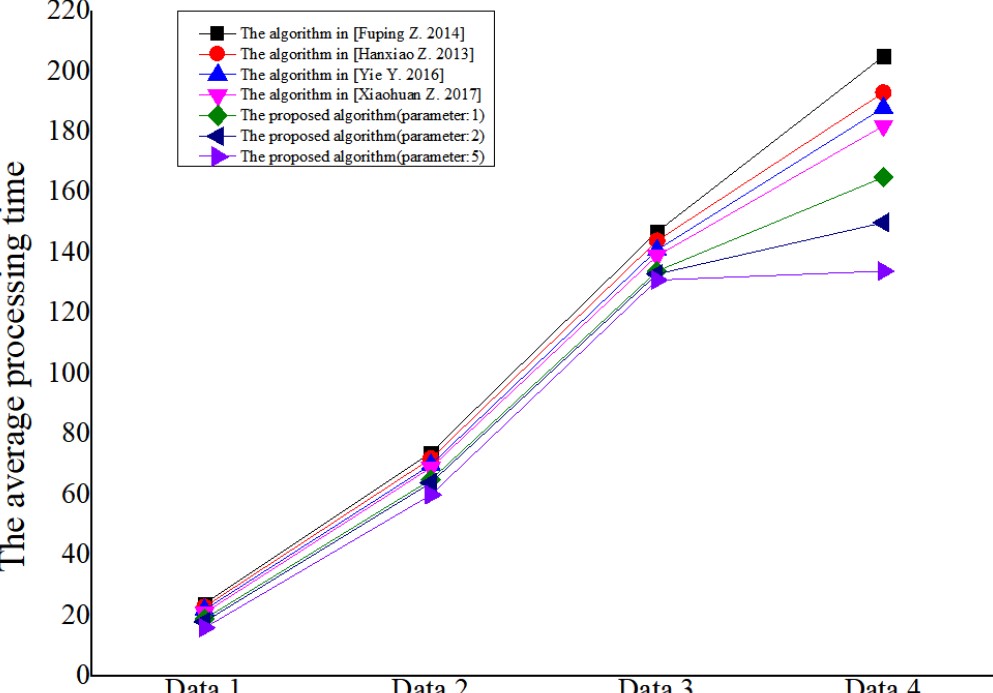

**Figure 15.** Comparison of the average scheduling results of each algorithm [15–18].

## 6. Conclusions

This study analyzes the shortcomings of the current two-workshop scheduling algorithm and proposes a scheduling algorithm for two-workshop production with the time-selective strategy and backtracking strategy. The algorithm is improved on the basis

of the algorithm proposed in [18] to avoid the algorithm falling into local optimization. At the same time, the solution space of the problem is further expanded. It has a significant relevance to further study of the integrated two-workshop scheduling problem. In addition, study of the multi-shop scheduling problem will form the basis of future work.

**Author Contributions:** Conceptualization, X.Z., Z.W.; Investigation, D.Z., T.X.; Methodology, X.Z., Z.W.; Project administration, X.Z., Z.W.; Software, X.Z., Z.W., D.Z.; validation, X.Z.; Supervision, X.Z.; Writing—original draft, X.Z., Z.W.; Writing—review & editing, X.Z., Z.W., D.Z., T.X.; supervision, X.Z.; funding acquisition, Z.W. and T.X. All authors have read and agreed to the published version of the manuscript.

**Funding:** This research was funded by The Professorial and Doctoral Scientific Research Foundation of Huizhou University (No.2019JB014, No.2018JB007), Guangdong Overseas Famous Teacher Project: Research on artificial intelligence and new generation intelligent manufacturing technology, 2021 practice teaching base project of integration of science, industry and education in Guangdong Undergraduate Universities: Huizhou information technology science, industry and education integration practice teaching base.

**Data Availability Statement:** There is no relevant authoritative verification data for the problem studied in this paper, and the usage data is randomly generated by the algorithm according to different situations.

**Acknowledgments:** We thank the editors and reviewers of the journal for their strong support of this manuscript.

**Conflicts of Interest:** The authors declare no conflict of interest.

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
