# Peer review of "Scheduling Algorithm for Two-Workshop Production with the Time-Selective Strategy and Backtracking Strategy"

_electronics, doi:10.3390/electronics11234049_

Round 1

Reviewer 1 Report

 Title: An Intelligent Scheduling Algorithm for Two-Workshop Production with Time-Selective Strategy

The structure of the paper

1.       Introduction

2.       Problem description and analysis

2.1.              Problem description

2.2.             Problem analysis

3.       Strategy design

4.The example analysis

5. Experimental comparison and analysis

 6. Conclusion

References (18)

The actual task of planning the work of two production shops is considered.

The problem statement is given.

A strategy for choosing the optimal solution is proposed.

An example of the application of the proposed approach is considered, and its analysis is carried out.

Remarks

1. It is not clear from the article whether this strategy can be used for more shops.

2. There is no analysis of the complexity of the proposed algorithm.

3. This class of problems belongs to combinatorial algorithms. No estimation of “sup” border.

4. The title of the paper mentions an intelligent algorithm. Not mentions in body of the paper.

Author Response

Authors’ Reply to Reviewers

Thanks the reviewers for giving us the helpful comments and suggestions, which helps us to improve the quality of the manuscript. We have formally acknowledged them in the revised manuscript under Acknowledgements section. In the revised version, we have incorporated their concerns and changes made in the revised manuscript are blue in color. The followings are our replies to the reviewers’ comments: the words in blue are the questions of the reviewers, and the words in black are our replies.

  1. It is not clear from the article whether this strategy can be used for more shops.

Reply to Question 1: This is a good question and we have made necessary changes. This part has been discussed in the revised manuscript.

  1. There is no analysis of the complexity of the proposed algorithm.

Reply to Question 2: This is a good question and we have made necessary changes. The complexity analysis of the proposed algorithm has been added to the revised manuscript.

  1. This class of problems belongs to combinatorial algorithms. No estimation of “sup” border.

Reply to Question 3: This is a good question and we have made necessary changes. The question has been revised in the manuscript.

The title of the paper mentions an intelligent algorithm. Not mentions in body of the paper.

Reply to Question4: This is a good question.Following the suggestions of the reviewer, we decided to change the title of the manuscript to: Scheduling Algorithm for Two-Workshop Production with the Time-Selective Strategy

Special thanks to you for your good comments.

We tried our best to improve the manuscript and made some changes in the script.These changes will not influence the content and framework of the paper.And here we did not list the changes but marked in blue in revised paper.

We appreciate for Editors/Reviewers’warm work earnestly,and hope that the correction will meet with approval.

Once again,thank you very much for your comments and suggestions.

Reviewer 2 Report

Furthermore, some drawbacks are still not be solved:

(1)   The English writing of the paper is required to be improved. Please check the manuscript carefully for typos and grammatical errors. The reviewer found some typos and grammatical errors within this manuscript, which have been excluded from my review. In addition, the English structure of the article, including punctuation, semicolon, and other structures, must be carefully reviewed.

(2) Future recommendations should be added to assist other researchers to extend the presented research analysis.

(3) The authors have to add some real applications to compare with other published articles.

(4) The authors have to compare their results with the other literature and present the advantages.

Author Response

Thanks the reviewers for giving us the helpful comments and suggestions, which helps us to improve the quality of the manuscript. We have formally acknowledged them in the revised manuscript under Acknowledgements section. In the revised version, we have incorporated their concerns and changes made in the revised manuscript are blue in color. The followings are our replies to the reviewers’ comments: the words in blue are the questions of the reviewers, and the words in black are our replies.

Point 1: The English writing of the paper is required to be improved. Please check the manuscript carefully for typos and grammatical errors. The reviewer found some typos and grammatical errors within this manuscript, which have been excluded from my review. In addition, the English structure of the article, including punctuation, semicolon, and other structures, must be carefully reviewed.

Response 1: This is a good question and we have made necessary changes. The English grammar of the full manuscript has been modified.

Point 2: Future recommendations should be added to assist other researchers to extend the presented research analysis.

Response 2: This is a good question, the outlook for future work has been added in the paper.

Point 3: The authors have to add some real applications to compare with other published articles.

Response 3:This is a good question and we have made necessary changes. The proposed algorithm has been compared with the existing better algorithm in published articles by running a real application.

Point 4: The authors have to compare their results with the other literature and present the advantages.

Response 4:This is a good question and we have made necessary changes. The advantages of this algorithm compared with other literatures have been supplemented in the manuscript.

Special thanks to you for your good comments.

We tried our best to improve the manuscript and made some changes in the script.These changes will not influence the content and framework of the paper.And here we did not list the changes but marked in blue in revised paper.

We appreciate for Editors/Reviewers’warm work earnestly,and hope that the correction will meet with approval.

Once again,thank you very much for your comments and suggestions.

Reviewer 3 Report

Manuscript Number: electronics-2047514-peer-review-v1

Title: An Intelligent Scheduling Algorithm for Two-Workshop Production with Time-Selective Strategy

This study focuses on solving the two-workshop integrated scheduling problem with the same device resources. The existing algorithms pay attention to the horizontal parallel processing of the process tree and ignore the tightness between the vertical serial processes. Authors proposed an Intelligent Scheduling Algorithm for two-Workshop Production with Time-Selective Strategy.

This paper does not have a satisfactory quality in the problem definition, solution method and outcomes.

- The paper is not a deep study in the sense of experimental study. Authors only listed simple, long and similar figures and tables after each other. Neither tables compared algorithms properly both analytically and computationally, nor they gave insights about the results in the paragraphs afterward. Let me give an example. Figures 11-19 can be grouped into one or two figures for the sake of simplicity. So, I disagree with the last paragraph of the paper saying that “the algorithm is improved on the basis of….”

- The paper is limited to a simple case study, and consequently it is not succeeded in a comprehensive study of (Two-Workshop) Production problems and giving a vision about big-picture comparison of algorithms. Again, the study is very much example based as figures and tables can show it.

- In Figures 3 and 4, workshop S2 has no job to present. Is it an error? If not, it may be helpful to remove workshop S2 from each figure.

- Tables mostly do not have valuable information. For example, Table 1 has occupied 4 pages of the paper without showing important information.

- What is the reason of having 14 definitions in Section 2.1. They are mostly trivial. Please keep only those which are necessary.

- The title of some papers in the reference list is lowercase, whereas the others are Capitalized each Word. This is also a case of inconsistency. Please make all of them lowercase for the sake of simplicity.

-Please avoid informal English. I see the use of “he’s, can’t, isn’t, aren’t, doesn’t, don’t, won’t”. Please avoid using informal form of these verbs. Examples:

Page 5: I won't repeat them here

-The paper benefits from a proofread. There are some typo errors in the paper.

Author Response

Thanks the reviewers for giving us the helpful comments and suggestions, which helps us to improve the quality of the manuscript. We have formally acknowledged them in the revised manuscript under Acknowledgements section. In the revised version, we have incorporated their concerns and changes made in the revised manuscript are blue in color. The followings are our replies to the reviewers’ comments: the words in blue are the questions of the reviewers, and the words in black are our replies.

Point 1: The paper is not a deep study in the sense of experimental study. Authors only listed simple, long and similar figures and tables after each other. Neither tables compared algorithms properly both analytically and computationally, nor they gave insights about the results in the paragraphs afterward. Let me give an example. Figures 11-19 can be grouped into one or two figures for the sake of simplicity. So, I disagree with the last paragraph of the paper saying that “the algorithm is improved on the basis of….”

Response 1: This is a good question and we have made necessary changes. In the previous manuscript, we did not realize this problem. Now, the experimental data has been redrawn and displayed, and the experimental data has also been further explained. In particular, Figure 11-19 has been redrawn for better reading.

Point 2: The paper is limited to a simple case study, and consequently it is not succeeded in a comprehensive study of (Two-Workshop) Production problems and giving a vision about big-picture comparison of algorithms. Again, the study is very much example based as figures and tables can show it.

Response 2: This is a good question, and we have made the necessary changes. Similar to the previous suggestion, this problem should be our negligence. It is necessary to explain that this study is not based on specific examples. In the original text, examples are introduced to explain the algorithm more clearly. The following experiment is not based on a certain example, but uses a large number of different data for comparison. After modification, this problem has been solved.

Point 3: In Figures 3 and 4, workshop S2 has no job to present. Is it an error? If not, it may be helpful to remove workshop S2 from each figure.

Response 3: Thank you very much for your careful reading. S2 in Figure 3 does not have a job. This is the result of algorithm arrangement, because migration time should be considered.

Point 4: Tables mostly do not have valuable information. For example, Table 1 has occupied 4 pages of the paper without showing important information.

Response 4: Table 1 specifically introduces the scheduling process of the proposed algorithm scheduling instance, which is helpful for readers to further understand the algorithm. If you still think it is unnecessary, I can delete it.

Point 5: What is the reason of having 14 definitions in Section 2.1. They are mostly trivial. Please keep only those which are necessary.

Response 5: This is a good question and we have made necessary changes. Now 14 definitions have been reduced to 8.

Point 6: The title of some papers in the reference list is lowercase, whereas the others are Capitalized each Word. This is also a case of inconsistency. Please make all of them lowercase for the sake of simplicity.

Response 6: Thank you very much for your careful reading. References have been revised to a consistent format.

Point 7: Please avoid informal English. I see the use of “he’s, can’t, isn’t, aren’t, doesn’t, don’t, won’t”. Please avoid using informal form of these verbs. Examples:

Page 5: I won't repeat them here

Response 7: This is a good question and we have made necessary changes. The English grammar of the full manuscript has been modified.

Point 8: The paper benefits from a proofread. There are some typo errors in the paper.

Response 8: This is a good question and we have made necessary changes.  The typo errors in the paper have been modified.

 Special thanks to you for your good comments.

We tried our best to improve the manuscript and made some changes in the script.These changes will not influence the content and framework of the paper.And here we did not list the changes but marked in blue in revised paper.

We appreciate for Editors/Reviewers’warm work earnestly,and hope that the correction will meet with approval.

Once again,thank you very much for your comments and suggestions.

Round 2

Reviewer 1 Report

All the comments are taken into account by the Authors.

Reviewer 2 Report

The revision can be accepted.

Reviewer 3 Report

I have had a quick look at the revision of the paper. The quality is now acceptable for publication in MDPI. My comments are answered (not all but mostly). 

The paper can be accepted as it is.